# Homozygous Inversion on Chromosome 13 Involving SGCG Detected by Short Read Whole Genome Sequencing in a Patient Suffering from Limb-Girdle Muscular Dystrophy

**DOI:** 10.3390/genes13101752

**Published:** 2022-09-28

**Authors:** Natalie Pluta, Sabine Hoffjan, Frederic Zimmer, Cornelia Köhler, Thomas Lücke, Jennifer Mohr, Matthias Vorgerd, Hoa Huu Phuc Nguyen, David Atlan, Beat Wolf, Ann-Kathrin Zaum, Simone Rost

**Affiliations:** 1Institute of Human Genetics, Biocenter, University of Würzburg, 97074 Würzburg, Germany; 2Department of Human Genetics, Ruhr-University Bochum, 44801 Bochum, Germany; 3Department of Neuropaediatrics, University Children’s Hospital, Ruhr-University Bochum, 44801 Bochum, Germany; 4Department of Neurology, Heimer Institute for Muscle Research, University Hospital Bergmannsheil, Ruhr-University Bochum, 44789 Bochum, Germany; 5Phenosystems SA, 1440 Braine le Chateau, Belgium; 6iCoSys, University of Applied Sciences Western Switzerland, 1700 Fribourg, Switzerland; 7Medical Genetics Center (MGZ), 80335 Munich, Germany

**Keywords:** inversion, sarcoglycanopathy, *SGCG*, whole genome sequencing (WGS), next generation sequencing (NGS), LGMDR5, muscle disease, genetic diagnostics

## Abstract

New techniques in molecular genetic diagnostics now allow for accurate diagnosis in a large proportion of patients with muscular diseases. Nevertheless, many patients remain unsolved, although the clinical history and/or the muscle biopsy give a clear indication of the involved genes. In many cases, there is a strong suspicion that the cause must lie in unexplored gene areas, such as deep-intronic or other non-coding regions. In order to find these changes, next-generation sequencing (NGS) methods are constantly evolving, making it possible to sequence entire genomes to reveal these previously uninvestigated regions. Here, we present a young woman who was strongly suspected of having a so far genetically unsolved sarcoglycanopathy based on her clinical history and muscle biopsy. Using short read whole genome sequencing (WGS), a homozygous inversion on chromosome 13 involving *SGCG* and *LINC00621* was detected. The breakpoint in intron 2 of *SGCG* led to the absence of γ-sarcoglycan, resulting in the manifestation of autosomal recessive limb-girdle muscular dystrophy 5 (LGMDR5) in the young woman.

## 1. Introduction

Sarcoglycanopathies belong to the group of limb-girdle muscular dystrophies (LGMDs), are inherited autosomal recessively, and result from homozygous or compound heterozygous pathogenic alterations in one of the four sarcoglycan genes, *SGCA*, *SGCB*, *SGCG*, or *SGCD* [1]. Although the phenotype can be similar within the sarcoglycanopathies, the clinical manifestation of a sarcoglycanopathy is highly variable both intra-familial and inter-familial [2]. Sarcoglycanopathies show progressive muscle weakness, increased serum creatine kinase (CK) levels and can occur in early childhood or adulthood. The muscle weakness manifests, among other things, by a positive Gowers sign, difficulty in climbing stairs, up to the loss of ambulation in adolescence or adulthood [1]. The sarcoglycanopathies were discovered by biochemical analysis of dystrophin, with variants in the corresponding gene being the genetic cause of Duchenne muscular dystrophy (DMD) or Becker muscular dystrophy (BMD) [1,2]. The four sarcoglycans (SGCs) α, β, γ, and δ form the so-called sarcoglycan complex and belong to the large dystrophin-glycoprotein complex (DGC), which is embedded in the sarcolemma of muscle cells and connects the extracellular matrix to the cytoskeleton. The associated sarcoglycan complex is responsible for stabilizing the sarcolemma [3].

Limb girdle muscular dystrophy 5 (LGMDR5, MIM: 253700, formerly LGMD2C), is one of the sarcoglycanopathies caused by pathogenic variations in *SGCG* [4]. The phenotype of these patients is very similar to that of DMD in terms of symptoms and even in the muscle biopsy, showing an exchange of muscle tissue into fatty and connective tissue on the one hand and variations in fiber size on the other hand [5]. Since two point mutations in *SGCG* (NM_000231.3:c.525del and c.848G>A) have been described as founder mutations and are found with above-average frequencies in patients [6], Sanger sequencing has been the first method of choice to detect single nucleotide variants (SNVs) in the coding regions of *SGCG* as the genetic cause of the disease.

In the meantime, next generation sequencing (NGS) technologies are used for the analysis of unclear LGMDs, either in the form of panel analysis in which mainly muscle-associated genes are included or in the form of whole exome sequencing, in order to adapt the search for variants in muscle-associated genes to the latest state of research [7]. As costs for whole genome sequencing (WGS) have been falling in recent years [8], this method is becoming more and more important in genetic diagnostics. The biggest advantage of WGS is that there is a chance to detect almost all possible genetic changes such as small and large deletions or duplications [9], coding and noncoding SNVs [10], small repeats [11], and inversions or other complex rearrangements [12], using just one method. The interpretation of these variants remains a major challenge, especially in the case of rare genetic diseases, as each patient may carry several potentially causative variants that have to be validated by further functional tests, which can be time-consuming and costly. However, if a variant in a known gene is detected in a patient that matches the phenotype, or which was suspected in advance but could not be solved by previous genetic diagnostics, there is a great chance of identifying the causative variant with the help of WGS [13].

Here we report on a young woman who suffers from muscle weakness since childhood. Previous genetic testing using NGS has failed to uncover the cause of her condition. The aim of this work was to find the genetic cause of the patient’s disease by using whole genome sequencing and to show that it is possible to detect structural variants by short-read sequencing.

## 2. Patient and Methods

### 2.1. Case Presentation

After an uneventful pregnancy, delivery, and early psychomotor development, the patient showed first signs of muscle weakness at the age of 7–8 years, when difficulties were observed during sports lessons. She first presented at the Children’s Hospital at 10 years of age with proximal muscle weakness, a positive Gowers sign and highly elevated CK levels (up to 15,000 U/L; norm < 150 U/L). A progressive form of muscular dystrophy was clinically suspected, but the family did not wish further analyses at this point. At the age of 15 years, the symptoms had further progressed: the patient was still able to walk unsupported, but only for small distances, and often needed help from family and friends. She showed marked scoliosis of the thoracical spine, a Trendelenburg gait, and atrophy of the biceps and ischiocrural muscles (strength 2/5) with less involvement of the extensor muscles (strength 4/5). EKG, echocardiography, and lung function tests gave normal results. The MRI scan of the lower extremities showed symmetrical, lipomatous dystrophic alterations, as well as edematous lesions, consistent with a dystrophic disorder, most likely LGMD. A muscle biopsy was taken from the vastus lateralis muscle. Immunostaining showed absence of γ-sarcoglycan, and likewise, γ-sarcoglycan was not detectable in Western blot analysis (Figure 1). Therefore, a γ-sarcoglycanopathy was highly suspected; however, Sanger sequencing of the *SGCG* gene as well as the other sarcoglycan genes did not reveal a pathogenic variant. A subsequent panel analysis including 65 genes known to be causative for muscular disorders also did not show a causative variant.

The patient’s parents and her brother are healthy with no signs of muscle disease. There was no consanguinity reported. The patient has given her consent for extended genetic diagnostics.

### 2.2. mRNA Analysis

Total mRNA from muscle tissue was extracted using the RNeasy Mini Kit (Qiagen, Hilden, Germany) according to the manufacturer’s instructions. The mRNA was transcribed into cDNA using the SuperScript II RT (ThermoFisher, Waltham, MA, USA). Some fragments were amplified directly from the extracted mRNA using the QIAGEN OneStep RT-PCR Kit (Qiagen, Hilden, Germany), others from the transcribed cDNA using standard PCR methods. PCR fragments were separated using the Bioanalyzer 2100 (Agilent, Santa Clara, CA, USA). Primers listed in Table 1 were used for PCRs from cDNA and mRNA.

### 2.3. Whole Genome Sequencing

Library preparation for whole genome sequencing was performed using the TruSeq DNA PCR-Free Library Prep. The sequence analysis was done by paired-end sequencing with the S1 Reagent Kit and 300 cycles on a NovaSeq 6000 Sequencing System (Illumina, San Diego, CA, USA). Sequence data alignment against the human reference genome GRCh38 (hg38) was performed using the generated fastq files and the software GensearchNGS (PhenoSystems SA, Wallonia, Belgium) [14] with the integrated aligner. The whole exome was evaluated with a virtual panel containing 1706 genes (list of genes available on request) assembled from various muscle associated terms from the Human Phenotype Ontology (HPO) [15]. Only missense, nonsense, small deletions/insertions, and splice variants (located up to 8 bp in the intron) were examined. In addition, analysis for copy number changes (CNVs) in the 1706 genes was performed using the integrated CNV tool of GensearchNGS.

The whole genome sequencing data were first used to investigate deep intronic variants. Structural variants were searched using the integrated so-called “anomaly scan” of GensearchNGS. This tool searches for areas where the sequence partially does not match the reference sequence and for areas where the paired reads are separated by a greater than average distance (above 300 bp). The software can distinguish between paired reads located within the same chromosome with a specified distance or on another chromosome. In order to perform a kind of “homozygosity mapping”, GensearchNGS was used to search for larger regions in which only homozygous variants are present.

Standard PCRs on genomic DNA followed by Sanger sequencing on an ABI 3130xl genetic analyzer (Thermo Fisher Scientific, Waltham, MA, USA) were performed to validate the detected anomalies. Sanger sequencing data were analyzed using Gensearch v.4.4 (PhenoSystems SA, Wallonia, Belgium). Breakpoints of the structural variant were verified in the patient and her parents by PCR with the primers SGCG-Int2-F, SGCG-Int2-R and LINC00621-Upstream-F, LINC00621-Upstream-R (Table 1). The parents were further screened for two variants detected in the daughter with Sanger sequencing, using the primers SGCG-Int1-g.23186799-F, SGCG-Int1-g.23186799-R, SGCG-Int1-g.23200194-F, and SGCG-Int1-g.23200194-R (Table 1).

### 2.4. Muscle Biopsy Analyses

Hematoxylin and eosin staining (HE) was performed on a muscle section of the patient. The patient’s muscle section was incubated in hematoxylin solution for 10 min to stain the nuclei, followed by incubation in eosin solution for 3 min. The section was placed in 70% and 90% ethanol for 20 s each, in 100% ethanol for 1 min, and finally in xylene for 3 min. For immunostaining, sections were fixed with acetone for 10 min and permeabilized with 0.1% Triton-X-100. After blocking with 2% BSA in PBS for 45 min, the three different antibodies against the sarcoglycan complex were added overnight in 2% BSA. Alexa-Fluor anti-mouse 488 at a dilution of 1:1000 in 2% BSA was applied to the sample after washing three times with PBS on an orbital shaker for 1 h. Before microscopy, the sections were again washed three times with PBS. Microscopy was performed using an Olympus IX83 microscope (Olympus Corporation, Tokyo, Japan).

For Western blotting, LDS-PAGE was performed followed by blotting with a wet blot to a nitrocellulose membrane overnight. The nitrocellulose membrane was blocked with 2% BSA in PBS for 1 h. The first antibody against γ-sarcoglycan was diluted 1:83.5 in 2% BSA in 0.05% PBS Tween and incubated overnight at 4 °C with the membrane. After washing, the second antibody against mouse peroxidase was diluted 1:10,000 and incubated with the membrane for 1 h. After washing three times with PBS Tween, the chemiluminescent substrate Radiance Q was added directly to the membrane and the proteins are visualized with the Azure 600 imager. For quantification, GAPDH staining is performed simultaneously and used as a reference protein for densitometry, which gives the relative expression of γ-sarcoglycan in the control and patient.

## 3. Results

Within routine diagnostics, analyses of the muscle gene panel (65 genes), MLPA of *SGCG,* and whole exome sequencing (1706 genes) did not yield relevant results for the patient’s disease. Because of the evidence from the muscle biopsy for the complete loss of γ-sarcoglycan (Figure 1), the initial genomic analysis focused on the evaluation of *SGCG*.

As a first result of WGS analysis, two homozygous variants in intron 1 of *SGCG* were detected, which according to ACMG guidelines [16] were classified as variants of uncertain significance: g.23200194G>A (rs140517497, MAF: 0.21%), and g.23186799A>C (rs147323292, MAF: 0.23%). Parental testing for the two detected variants showed that the parents are heterozygous carriers of both variants. Therefore, homozygosity mapping with GensearchNGS was performed and showed a large homozygous region of about 5.5 Mb on chromosome 13 including *SGCG* (Figure 2). The next step was to analyze *SGCG* for anomalies, which revealed a conspicuous region in intron 2 around g.23223194 with many reads whose paired reads are located above average with a nearly identical distance of about 350 kb apart (Figure 3a). These reads also contain many homozygous and heterozygous appearing variants, some with over >70% allele frequency in the population and some with a MAF below 0.1%.

Jumping to the location of those paired reads in GensearchNGS showed that these are all located downstream of the gene *LINC00621* around g.22869698. At this position there are also reads with the same distance of approximately 350 kb as the conspicuous reads in *SGCG* (Figure 3b). It can be seen that in both areas of the long-distance reads (upstream of *LINC00621* and in intron 2 of *SGCG)*, there are AluY repeats. These become visible by an embedded bed.file of the RepeatMasker (Smit, AFA, Hubley, R & Green, P. *RepeatMasker Open-4.0*. 2013–2015 http://www.repeatmasker.org, accessed on 25 May 2021). In order to find out what kind of genetic alteration is responsible for this abnormality, PCR followed by Sanger sequencing around the conspicuous regions was carried out. In the patient, PCR with primers LINC00621-Upstream-F/LINC00621-Upstream-R and SGCG-Int2-F/SGCG-Int2-R failed to amplify any products, whereas in the control, products of the expected sizes of 1030 bp (*LINC00621*) and 1577 bp were detected. Combining the primer pairs LINC00621-Upstream-F and SGCG-Int2-R or the other way around, fragments of the desired sizes were obtained only in the patient. Sanger sequencing of these fragments revealed that the orientation of the two genes (*SGCG* and *LINC00621*) was opposite and narrowed down the breakpoint of an assumed inversion to a sequence consisting of 19 bp (Figure 4b). Due to the 87% concordance of the two *AluY* repeats involved, a more precise statement about the breakpoint could not be made. The divergence of the sequences marked in the electropherogram shows where the sequence of *LINC00621* ends and the sequence of *SGCG* starts. With these results and the validation by PCR, the inversion scheme is as shown in Figure 4a. To verify whether the inversion was inherited from both parents, the region of the suspected inversion was also amplified in the DNA of the parents. It showed fragments in both parents with primers LINC00621-Upstream-F/LINC00621-Upstream-R and SGCG-Int2-F/SGCG-Int2-R as well as with primers LINC00621-Upstream-F/SGCG-Int2-R.

Amplification of the patient’s cDNA revealed no product for the *SGCG* transcript and for *LINC00621* in any case, whereas the control showed products of the expected sizes of 398 bp, using primers SGCG-cDNA-Ex1-F and SGCG-cDNA-Ex4-R (Figure 5a) and 783 bp for the fragment with primers LINC00621-cDNA-Ex2-F/LINC00621-cDNA-Ex4-R (Figure 5b). PCR analyses of the patient’s cDNA for *DYSF* served as RNA quality control and showed that the fragments could be amplified in the expected size of 457 bp for DYSF-cDNA-Ex10-F and DYSF-cDNA-Ex15-R.

## 4. Discussion

Here, we present a young woman suffering from progressive proximal muscle weakness and strongly increased serum creatine kinase levels. By previous routine diagnostics that included MLPA analysis, panel sequencing and whole exome sequencing, it was not possible to identify the patient’s genetic cause. Several facts suggested that the genetic cause of the myopathy of this patient must lie within the *SGCG* gene: First, the muscle biopsy showed a lack of γ-sarcoglycan. Next, it was striking that in the genome data, *SGCG* is located in a larger homozygous region on chromosome 13, and its transcript could not be amplified from a muscle RNA sample. Whole genome analysis of this case revealed a homozygous inversion on chromosome 13 with a breakpoint downstream of *LINC00621* and within intron 2 of *SGCG*. The formation of the inversion can be explained by the presence of two highly homologous *AluY* repeats in *SGCG* and *LINC00621*. The inversion in this case is the genetic cause of the patient’s muscle disease as the disruption of the *SGCG* gene prevents its expression. Disruption of a gene by inversion and thus the manifestation of the disease is previously described for several genetic disorders, e.g., in Hunter syndrome or hemophilia A [17,18].

Since the search for SNVs and CNVs in the whole genome data was unsuccessful, the so-called anomaly scanner of GensearchNGS was used to search for structural changes in the *SGCG* gene. Structural variations (e.g., deletions, duplications, translocations, inversions) can be seen on the one hand when the majority of a read matches the reference sequence but a smaller part deviates from it and contains a different “foreign” sequence. On the other hand, structural variants can be seen when paired reads are more than 300 bp apart from each other, which are called “bad pairs” in the software used for data analysis. This is conspicuous, due to the fact that the enrichment method produces fragments and hence, reads that are no longer than 600 bp. GensearchNGS can display the distance of a read and its paired read or indicate whether the paired reads are located on the same or on a different chromosome.

Structural changes such as inversions have long been known as the underlying mechanism for genetic diseases. The best-known example is the intron 22 inversion in hemophilia A, which is one of the most common genetic causes of this disease [18]. However, also in muscle diseases, especially in *DMD,* intragenic inversions could be detected as a genetic cause for the disease. The detection of small inversions (that could not be seen in a conventional karyogram) is usually done by Southern Blot, RNA, or cDNA analysis followed by stepwise PCR to narrow down the breakpoints [17,19,20,21]. Routine diagnostic methods such as MLPA or array CGH are only conditionally suitable and can only provide clues for underlying inversions [20]. It is now possible to detect such complex structural changes and capture them in their entirety using the latest sequencing methods, i.e., long read sequencing [22]. In the here presented patient, routine diagnostic methods, especially sequencing of coding sequences, did not give any clue to the genetic cause because the breakpoints of the inversion are located in areas (intronic and downstream) that are not covered by these methods. Additionally, it was not possible to detect the genetic cause by RNA analysis, because *SGCG* could not be amplified from cDNA or directly from muscular RNA. It has been described that gene expression can be disturbed by inversions, since regulatory regions of a gene are separated from the coding regions by the break [23]. This could be the reason why *SGCG* could not be amplified in the patient’s RNA. The location of the *SGCG* promoter is not yet sufficiently described in humans, but it is in the orthologous gene in mice [24]. Comparing the sequences of the mouse promoter with the sequence in humans, the promoter could be located in the region including the untranslated exon 1 of *SGCG*. Since the breakpoint of the inversion described here is located downstream of the promoter in intron 2, the regulatory region would be separated from the coding sequence. 

So-called Alu elements belong to the mobile elements in the human genome and account for about 11% of the entire genome [25]. Due to their high sequence similarity and their frequent occurrence (about every 4 kb in the human genome), they can be the cause of recombination in the genome and thus cause structural changes such as inversions. The formation of inversions due to Alu elements results either from the formation of secondary structures or from the formation of a fragile site where a double strand break can occur. AluY repeats are often involved in these inversions caused by recombination because they are younger than other Alu repeats and thus have higher sequence similarity [25,26,27]. The reason for the inversion in the here presented case is the presence of AluY repeats within intron 2 of *SGCG* and downstream of *LINC00621*, with a high similarity of 87%.

The breakpoint in intron 2 of *SGCG* and thus the presumably prevented formation of the mRNA of *SGCG* is the cause of the patient’s muscle disease. The question of whether the breakpoint downstream of *LINC00621* has an effect on the patient’s phenotype remains unclear. Long intergenic noncoding (linc) RNAs are transcribed RNAs that are longer than 200 bp but do not encode any protein [28]. The first known lincRNA that plays an important role in the human genome by compensating for the dosage of two X chromosomes in females through X inactivation is *Xist* (X-inactive specific transcript) [29]. So far, however, only a small fraction of lincRNAs is known to have a potential function. The known functions of lincRNAs are, for example, the participation in chromatin remodeling or the regulation of translation and transcription [30]. As far as we know, the patient has no other symptoms apart from her muscle disease. Whether the missing *LINC00621*-RNA in the patient has an effect on the phenotype that may not have been recognized yet cannot be finally evaluated due to the lack of knowledge about *LINC00621* in particular and also about lincRNAs in general.

This case clearly shows that with an appropriate software that includes various tools, such as scanning for far apart paired reads, it is possible to find such large and complex structural changes even in short read whole genome sequencing data. This also requires a good clinical history of a patient, which can further narrow down the area of interest. While inversions have been described as a rare cause of DMD [19,20,21,31] as well as other well-characterized hereditary disorders such as hemophilia [18] or familial cerebral cavernous malformation [32], inversions as genetic alterations have not yet been described in the literature in *SGCG* as well as in all other sarcoglycan genes to the best of our knowledge. Hence, this is the first published case of a homozygous inversion involving *SGCG* detected by short read WGS in a patient suffering from LGMD.

## Figures and Tables

**Figure 1 genes-13-01752-f001:**
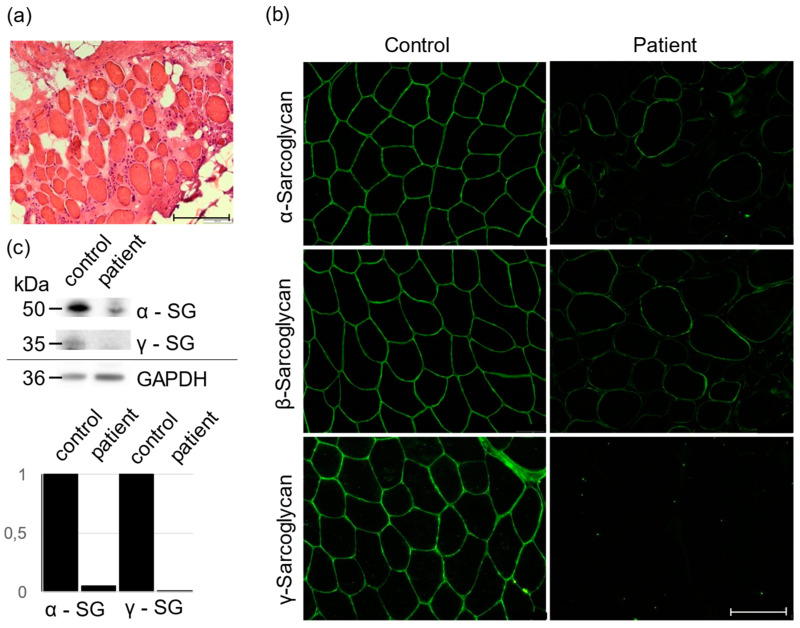
Haematoxylin and Eosin (HE) (**a**) and immunofluorescence staining with monoclonal antibodies (α-SG Novocastra/Leica, β- and γ-SG/Novocastra) (**b**) of serial transverse cryosections. Muscle specimens from the patient (**a**,**b**) and an unaffected control (**b**). In the patient, α- and β-SG were reduced, and γ-SG was completely absent. Scale bars, 100 μm. (**c**) Western Blot data with quantification demonstrated drastically reduction of α-SG and absence of γ-SG in the patient compared to control.

**Figure 2 genes-13-01752-f002:**
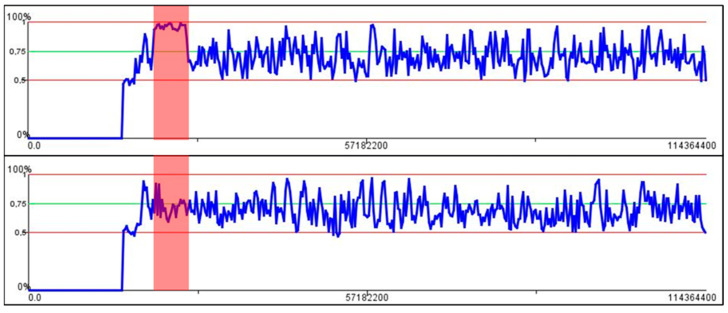
Homozygous area on chromosome 13 of the patient (**above**) and the same region in a control patient (**bottom**) are highlighted in red. In the patient, a plateau at 100% can be seen, which means that in this range all detected variants are present with a frequency of 100% (=homozygous).

**Figure 3 genes-13-01752-f003:**
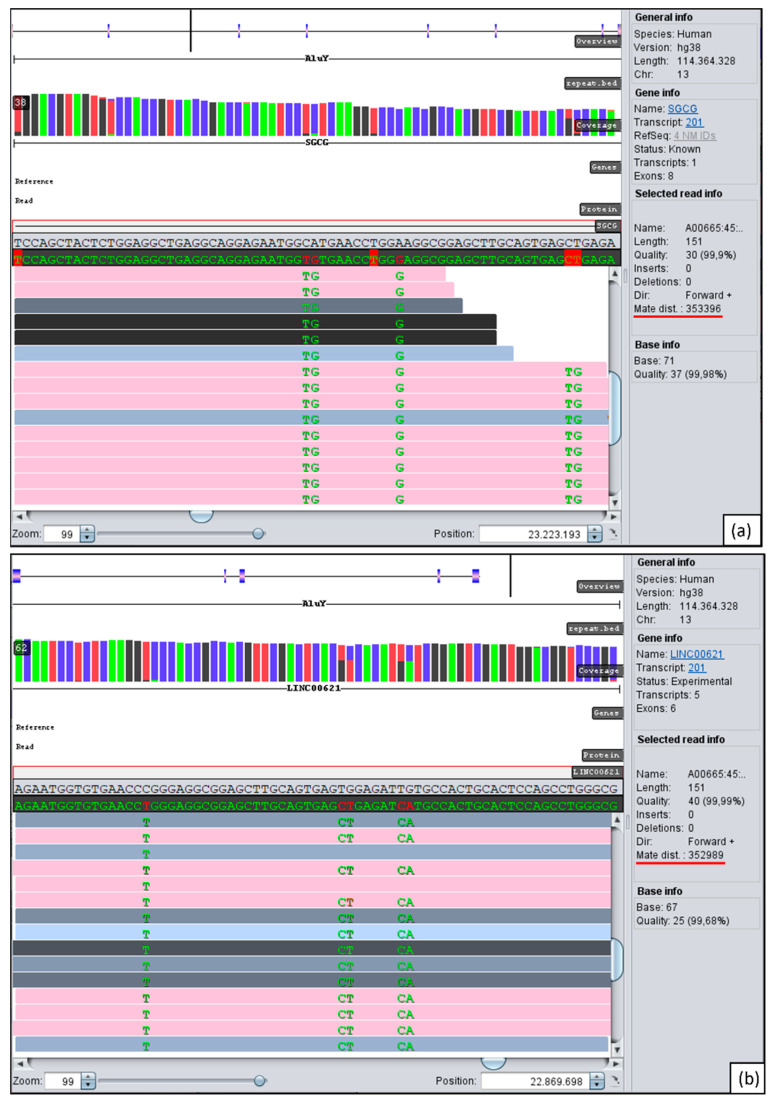
Parts of WGS reads on chromosome 13: (**a**) Region in intron 2 of *SGCG* around g.23223193. (**b**) Region upstream of *LINC00621* around g.22869698. The pink lines show reads whose pairs are more than 300 bp apart. The first line provides an overview of the gene with exons depicted as blue rectangles and showing the location of the visible reads within this gene as a black vertical stroke. The reference sequence is the grey backgrounded sequence. The distance of about 350,000 bp of the paired read can be seen at “Mate dist.” (underlined in red). The second line from the top shows the position of the two AluY repeats in the respective regions. Variants shown as green letters are due to differences between the AluY sequences.

**Figure 4 genes-13-01752-f004:**
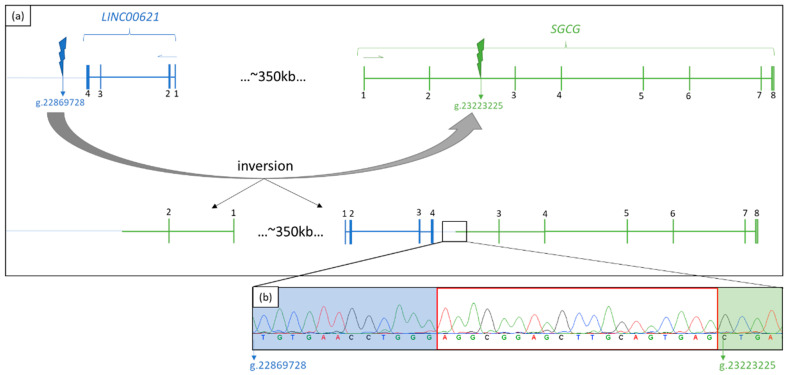
(**a**) Scheme of inversion. The exon numbering is listed in black numbers and the reading direction of the genes is indicated with a blue (*LINC00621*) or green (*SGCG*) arrow. (**b**) Sanger sequencing of the fragment LINC00621-Upstream-F and SGCG-Int2-R. Marked in blue is the unique sequence of *LINC00621* upstream and marked in green is the unique sequence of *SGCG* intron 2. The area in which the breakpoint can be located is outlined in red.

**Figure 5 genes-13-01752-f005:**
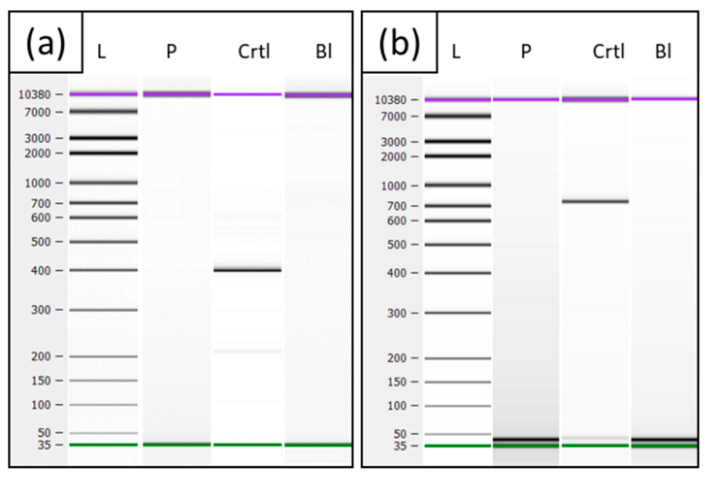
Electrophoresis of cDNA (**a**) cDNA PCR fragment of *SGCG* Ex1-4 from the patient (P) a healthy control (Ctrl) and a blank value (Bl). (**b**) cDNA PCR fragment of *LINC00621* Ex2-4 from patient (P), a healthy control (Ctrl), and blank value (Bl).

**Table 1 genes-13-01752-t001:** Primers used for PCR and Sanger sequencing.

Primer Name	Sequence	Used Transcript (RefSeq)
SGCG-cDNA-Ex1-F	5′-ATTCGCCAGTGTGCTTTTCT-3′	NM_000231.2
SGCG-cDNA-Ex4-R	5′-TGACCTCCCCTTCTGAGTTG-3′	NM_000231.2
DYSF-cDNA-Ex10-F	5′-GGGCACCATTTACAGAGAGC-3′	NM_003494.3
DYSF-cDNA-Ex15-R	5′-TTCGCACATGGAGGGAAA-3′	NM_003494.3
SACS-cDNA-Ex1-F	5′-TGGAGTACACATTTAAAGGAAGCA-3′	NM_014363.5
SACS-cDNA-Ex5-R	5′-AAGAATCTGTCCTCCTTCTGGA-3′	NM_014363.5
LINC00621-cDNA-Ex2-F	5′-CCAAGTCTGGTCAACCCAAT-3′	XR_001732839.1
LINC00621-cDNA-Ex4-R	5′-CTTTTTCCCATGTGAGGACAC-3′	XR_001732839.1
SGCG-Int2-F	5′-GGTATCTAATTCAATCAGCACTTTTT-3′	NM_000231.2
SGCG-Int2-R	5′-GCATGACACATACATGCCTGA-3′	NM_000231.2
LINC00621-Upstream-F	5′-TTGAAGTAAATGTTTTCTAACCTGTCA-3′	XR_001732839.1
LINC00621-Upstream-R	5′-TGATTTCAGTTTTATGTGGCATT-3′	XR_001732839.1
SGCG-Int1-g.23186799-F	5′-AAGTATCCCCCATCCTCACA-3′	NM_000231.2
SGCG-Int1-g.23186799-R	5′-GGCCAGCATCACTATCCAAG-3′	NM_000231.2
SGCG-Int1-g.23200194-F	5′-AGCCTCACCAGACTGAAGGA-3′	NM_000231.2
SGCG-Int1-g.23200194-R	5′-CCTCCAGGGTTTCAAGTGAG-3′	NM_000231.2

## Data Availability

As the data presented in this study are sequencing results of human samples, the data are not publicly available due to personal data protection.

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
