# Peer review of "Homozygous Inversion on Chromosome 13 Involving SGCG Detected by Short Read Whole Genome Sequencing in a Patient Suffering from Limb-Girdle Muscular Dystrophy"

_genes, 2022, doi:10.3390/genes13101752_

Round 1

Reviewer 1 Report

Pluta et al. describe the first case of homozygous inversion disrupting the SGCG gene and leading to autosomal recessive limb-girdle muscular dystrophy 5. The thorough description of the diagnostic process is not novel, but serves as a good reminder of different approaches needed to detect genetic abnormalities. The manuscript will be better ones the following concerns are addressed:

Major comment:
Typically, a single case report is not sufficient for a publication. However, inversions are rarely described in LGMD patients. I think this manuscript would be improved by an additional paragraph in the Discussion section reviewing the previously described cases of inversions in genes causing muscular disease.

Minor comments:
1. Please include the antibody references for the western blots (same as for immunohistochemistry?).

2. It is not clear how many different experiments were used to generate the bar graph with the quantification of western blot. The absence of gamma-SG is already clear from the blot, so there is no need to include a bar graph (unless proper quantification with several experiments and error bars has been performed).

3. Page 9 “outine diagnostic methods like MLPA or array CGH are hardly or only conditionally suitable to detect inversions”
It is not clear what authors mean by “conditionally suitable”, nor how an array-CGH can detect an inversion…

4. Page 9 “promoter could be located in the region of the untranslated exon 1 of SGCG
The statement is confusing, since “in the region” can be misinterpreted as “within exon1”.

5. Can you provide more information about the two ALU elements underlying the insertion – are they present in the reference genome (seem to be the case judging from the Figure 3)? If not, please provide the frequency in the general population (can be estimated from the gnomAD SVs v2.1).

6. Figure 3 is confusing – several tracks do not seem to be important for the manuscript. What does “Overview” track show? The white space between the reads and the position coverage is not necessary. Also, please show a more zoomed out view of the reads.

7. Page5 The authors state “These reads also contain many homozygous and heterozygous appearing variants, some with over >70% allele frequency in the population 163 and some with a MAF below 0.1%.”  However, they do not clarify for the reader that these “variants” are due to differences between the two AluY sequences. Please explain this in the text and in the Figure 3 legend.

Author Response

Pluta et al. describe the first case of homozygous inversion disrupting the SGCG gene and leading to autosomal recessive limb-girdle muscular dystrophy 5. The thorough description of the diagnostic process is not novel, but serves as a good reminder of different approaches needed to detect genetic abnormalities. The manuscript will be better ones the following concerns are addressed:

Major comment:
Typically, a single case report is not sufficient for a publication. However, inversions are rarely described in LGMD patients. I think this manuscript would be improved by an additional paragraph in the Discussion section reviewing the previously described cases of inversions in genes causing muscular disease.

Published cases about muscular disease caused by inversions are very sparse, there are reported cases of DMD which we already discussed, and we have also added additional examples to the discussion (see last paragraph). But other cases are scarce. Therefore, this case as the first published case of a sarcoglycanopathy due to an inversion is so important.

Minor comments:
1. Please include the antibody references for the western blots (same as for immunohistochemistry?).

We added methods descriptions for the muscle biopsy analyses, including Western blot (with quantification) and immunofluorescence. Antibody references are already given in the figure legend of Figure 1.

  1. It is not clear how many different experiments were used to generate the bar graph with the quantification of western blot. The absence of gamma-SG is already clear from the blot, so there is no need to include a bar graph (unless proper quantification with several experiments and error bars has been performed).

With the bar graph we want to visualize the quantification of the reduced gamma SG relative to the internal control (GAPDH) in addition to the Western blot. This allows the reader a quick orientation about the drastic decrease of the primary protein defect. This quantification only shows the Western blot results and therefore has no error bars.

  1. Page 9 “outine diagnostic methods like MLPA or array CGH are hardly or only conditionally suitable to detect inversions”
    It is not clear what authors mean by “conditionally suitable”, nor how an array-CGH can detect an inversion…

We wanted to point out that inversions are usually not detected by array or MLPA, unless they are accompanied by a loss or gain of genetic material in the breakpoint regions (and for MLPA, only if the copy number alterations are located within exons). In those cases, they can provide clues for the existence of structural variations (such as inversion) as described by Flanigan et al. We changed the sentence to: “Routine diagnostic methods like MLPA or array CGH are only conditionally suitable and can only provide clues for underlying inversions [20]”

  1. Page 9 “promoter could be located in the region of the untranslated exon 1 of SGCG
    The statement is confusing, since “in the region” can be misinterpreted as “within exon1”.

We changed the wording to „to the region including the untranslated exon 1 of SGCG.

  1. Can you provide more information about the two ALU elements underlying the insertion – are they present in the reference genome (seem to be the case judging from the Figure 3)? If not, please provide the frequency in the general population (can be estimated from the gnomAD SVs v2.1).

The ALU elements are present in the reference genome (hg38) chr13:23222997-23223298 and chr13:22869616-22869929 according to the UCSC browser.

  1. Figure 3 is confusing – several tracks do not seem to be important for the manuscript. What does “Overview” track show? The white space between the reads and the position coverage is not necessary. Also, please show a more zoomed out view of the reads.

The „Overview“ track provides information which location in the gene is depicted. We included this information in the figure legend. The tracks are depicted as seen in the program (GensearchNGS); as we did not want to temper with the data, we showed the image as provided.

  1. Page5 The authors state “These reads also contain many homozygous and heterozygous appearing variants, some with over >70% allele frequency in the population 163 and some with a MAF below 0.1%.”  However, they do not clarify for the reader that these “variants” are due to differences between the two AluY sequences. Please explain this in the text and in the Figure 3 legend.

We added this information in the figure legend of Figure 3. In the text, the role oft he AluY sequences is already explained in the paragraph following the abovementioned sentence.

Reviewer 2 Report

Pluta N et al., report a genetic mutation analysis of a young woman patient suspected of gamma sarcoglycanopathy. The introduction and discussion sections are well described.  Methods need some improvements and results need minor changes.

Methods: 1. Needs to describe the methods for all the presented results. Eg., Western blot, Immunofluorescence, H&E.

2. Describe how Western blot is quantified.

Results: 1. Why is the data for lines "187 & 189" not shown? If possible both results could be combined in figure 4.

2. Figure 5C is not needed or can be included in the supplemental section.

Minor spell correction: Line 48 as "girdle"

Author Response

Pluta N et al., report a genetic mutation analysis of a young woman patient suspected of gamma sarcoglycanopathy. The introduction and discussion sections are well described.  Methods need some improvements and results need minor changes.

Methods: 1. Needs to describe the methods for all the presented results. Eg., Western blot, Immunofluorescence, H&E.

  1. Describe how Western blot is quantified.

We added methods descriptions for the muscle biopsy analyses, including Western blot (with quantification) and immunofluorescence.

Results: 1. Why is the data for lines "187 & 189" not shown? If possible both results could be combined in figure 4.

Because of a low resolution of the (analogous) PCR fotos and because these data are in our opinion not necessary to understand the nature of the detected inversion, we decided not to include these pictures in the manuscript.

  1. Figure 5C is not needed or can be included in the supplemental section.

We deleted figure 5C.

Minor spell correction: Line 48 as "girdle"

We corrected this typing error.

Reviewer 3 Report

The authors have presented a nice well-written manuscript presenting the description of a young woman with progressive limb-girdle muscle weakness without associated cardiomyopathy and with clinical features and serum muscle biomarkers suggestive of a muscular dystrophy phenotype, especially limb-girdle muscular dystrophy. As immunostaining demonstrated absence of gamma-sarcoglycan in muscle biopsy, LGMDR5 was suspected. After a nice strategy to perform genetic study by means of whole genome sequencing, the authors have evinced homozygous inversion involving the SGCG gene. This case report brings a new description of a rare genetic mechanism involved in the occurrence of LGMDR5, as inherited inversions are rarely included as a potential cause of homozygous variants in autosomal recessive neuromuscular disorders. This manuscript emphasizes the need of detailed genetic evaluation of neuromuscular disease patients with a highly suggestive etiological basis and initial negative gene sequencing. 

Author Response

We thank the reviewer for this supportive review and do not see any points that need to be addressed.